# Frequency of co-seropositivities for certain pathogens and their relationship with clinical and histopathological changes and parasite load in dogs infected with *Leishmania infantum*

Valéria da Costa Oliveira[1©], Artur Augusto Velho Mendes Junior[1‡], Luiz Claudio Ferreira[1‡], Tatiana Machado Quinates Calvet[1‡], Shanna Araujo dos Santos[1‡], Fabiano Borges Figueiredo[2‡], Monique Paiva Campos[2‡], Francisco das Chagas de Carvalho Rodrigues[1‡], Raquel de Vasconcellos Carvalhaes de Oliveira[1‡], Elba Regina Sampaio de Lemos[3‡], Tatiana Rozental[3‡], Raphael Gomes da Silva[3‡], Maria Regina Reis Amendoeira[3‡], Rayane Teles-de-Freitas[3‡], Rafaela Vieira Bruno[3,4‡], Fernanda Nazaré Morgado[3‡], Luciana de Freitas Campos Miranda[1‡], Rodrigo Caldas Menezes[1©]*

1 Instituto Nacional de Infectologia Evandro Chagas, Fundação Oswaldo Cruz, Rio de Janeiro, Brazil,
2 Instituto Carlos Chagas, Fundação Oswaldo Cruz, Curitiba, Paraná, Brazil, 3 Instituto Oswaldo Cruz,
Fundação Oswaldo Cruz, Rio de Janeiro, Brazil, 4 Instituto Nacional de Ciência e Tecnologia em
Entomologia Molecular (INCT-EM)/CNPq, Rio de Janeiro, Brazil

© These authors contributed equally to this work.
‡ These authors also contributed equally to this work.
* rodrigo.menezes@ini.fiocruz.br

## Abstract

In canine leishmaniosis caused by the protozoan *Leishmania infantum*, little is known about how co-infections with or co-seropositivities for other pathogens can influence aggravation of this disease. Therefore, the objectives of this study were to evaluate the frequency of co-infections with or co-seropositivities for certain pathogens in dogs seropositive for *L. infantum* and their relationship with clinical signs, histological changes and *L. infantum* load. Sixty-six *L. infantum*-seropositive dogs were submitted to clinical examination, collection of blood and bone marrow, culling, and necropsy. Antibodies against *Anaplasma* spp., *Borrelia burgdorferi* sensu lato, *Ehrlichia* spp. and *Toxoplasma gondii* and *Dirofilaria immitis* antigens were investigated in serum. Samples from different tissues were submitted to histopathology and immunohistochemistry for the detection of *Leishmania* spp. and *T. gondii*. Quantitative real-time PCR was used to assess the *L. infantum* load in spleen samples. For detection of *Coxiella burnetii*, conventional PCR and nested PCR were performed using bone marrow samples. All 66 dogs tested positive for *L. infantum* by qPCR and/or culture. Fifty dogs (76%) were co-seropositive for at least one pathogen: *T. gondii* (59%), *Ehrlichia* spp., (41%), and *Anaplasma* spp. (18%). Clinical signs were observed in 15 (94%) dogs monoinfected with *L. infantum* and in 45 (90%) dogs co-seropositive for certain pathogens. The *L. infantum* load in spleen and skin did not differ significantly between monoinfected and co-seropositive dogs. The number of inflammatory cells was higher in the spleen, lung and mammary gland of co-seropositive dogs and in the mitral valve of monoinfected dogs. These results suggest that dogs infected with *L. infantum* and co-seropositive for certain

**Data Availability Statement:** All relevant data are within the manuscript and its Supporting information files.

**Funding:** This study was supported by the state funding agency Fundação Carlos Chagas Filho de Amparo à Pesquisa do Estado do Rio de Janeiro (Grants: CNE E-26/203.069/2016 and TCT E-26/202.561/2017), http://www.faperj.br/, and by Coordenação de Aperfeiçoamento de Pessoal de Nível Superior (CAPES), Brazil (Finance Code 001), https://www.capes.gov.br/. FBF, ERSL and RCM are recipients of productivity fellowships from Conselho Nacional de Desenvolvimento Científico e Tecnológico (CNPq), Brazil, http://www.cnpq.br/. The funders had no role in study design, data collection and analysis, decision to publish, or preparation of the manuscript.

**Competing interests:** The authors have declared that no competing interests exist.

pathogens are common in the region studied. However, co-seropositivities for certain pathogens did not aggravate clinical signs or *L. infantum* load, although they were associated with a more intense inflammatory reaction in some organs.

## Introduction

In Brazil, visceral leishmaniosis (VL) caused by the protozoan *Leishmania* (*Leishmania*) *infantum* is a zoonosis of important public health concern. From 1990 to 2019, there were 93,614 confirmed human cases of VL in Brazil, with an average of 3,120 new cases per year [1]. The main vector responsible for the transmission of this agent in the country is the sand fly *Lutzomyia longipalpis* [2]. In urban areas, the dog (*Canis familiaris*) is the main reservoir of *L. infantum* and a source of vector infection [3].

Co-infections with or co-seropositivities for other pathogens are common in dogs infected with *L. infantum*. These pathogens include the protozoa *Babesia* spp., *Cystoisospora* spp., *Giardia duodenalis*, *Hepatozoon canis*, *Neospora caninum*, *Toxoplasma gondii* and *Trypanosoma* spp. and bacteria such as *Anaplasma* spp., *Bartonella henselae*, *Borrelia burgdorferi*, *Ehrlichia canis* and *Rickettsia* spp., as well as the helminths *Acanthocheilonema reconditum*, *Ancylostoma caninum*, *A. braziliense*, *Dioctophyme renale*, *Dipylidium caninum*, *Dirofilaria immitis*, *Toxocara canis* and *Trichuris vulpis* [4–13]. The presence of co-infections in dogs with leishmaniosis caused by *L. infantum* can aggravate the disease and increase animal mortality [7, 10, 13, 14]. However, studies that evaluate the parasite load and histological changes in different organs together with clinical changes in co-infections with or co-seropositivities for *L. infantum* in dogs are scarce. Such investigations would be important to determine whether dogs infected with *L. infantum* and co-infected with or co-seropositive for certain pathogens have a greater potential for transmission of *L. infantum*, and to subsequently develop preventive and control measures for the involved agents. Within this context, the objectives of this study were to evaluate the frequency of co-infections with or co-seropositivities for certain pathogens in dogs infected with *L. infantum* and their relationship with clinical signs, histological changes and *L. infantum* load.

## Materials and methods

### Dog population

This is a descriptive study of 66 dogs (38 males and 28 females) that tested seropositive for *L. infantum* during the period from August 2016 to January 2019. Fifty-six of these dogs were mongrel, three were Pinschers, two were Pit Bulls, two were Rottweilers, one was a Dachshund, one was a Poodle, and one was a Chow-Chow. The age of the dogs ranged from 1 to 7 years in 58 (87.9%) animals, seven (10.6%) were older than 7 years, and one (1.5%) was less than 11 months old. All dogs were from the town of Barra Mansa, state of Rio de Janeiro, Brazil. This town is located in the south of the state (22˚32'25.19" S and 44˚10'35.33" W) and is an endemic area for human and canine leishmaniosis caused by *L. infantum* [15, 16]. All animals had owners and tested seropositive for anti-*Leishmania* antibodies by the rapid dual-path platform (DPP®) assay (TR DPP®) [17] and by enzyme immunoassay (ELISA) [18]. Both serological tests are produced by BioManguinhos (Fiocruz, Rio de Janeiro, Brazil). These tests were performed by public health services participating in the VL surveillance and control program of the state of Rio de Janeiro, with permission of the owners. Since they tested positive, the

dogs were sent by the Municipal Health Department of Barra Mansa to be culled at the Evandro Chagas National Institute of Infectious Diseases (INI-Fiocruz). Culling was performed according to the recommendations of the Brazilian Ministry of Health for the control of VL [2] and the owners provided signed consent for culling. The dogs were not housed for any period of time prior to culling. The sample size included all dogs of the study population, i.e., all dogs that tested seropositive for *L. infantum* in the town of Barra Mansa and that were sent to INI-Fiocruz during the study period.

## Sample collection

Immediately after arrival at INI-Fiocruz, the dogs were restrained mechanically and submitted to clinical evaluation, including inspection of behavior, alertness, posture, gait, skin, and oral and ocular mucosae, as well as palpation of the superficial lymph nodes and organs. The following clinical signs of canine leishmaniosis caused by *L. infantum* (CanL) were considered: thinness or cachexia; diffuse or localized alopecia; apathy; cutaneous lesions such as ulcers and desquamation; onychogryphosis; enlargement of the superficial lymph nodes, liver or spleen on palpation; keratoconjunctivitis; pale ocular or oral mucosae, and skeletal muscle atrophy [19, 20]. The animals were divided into three groups according to the clinical signs of CanL: no clinical signs, few clinical signs (up to three clinical signs), and multiple clinical signs (more than three clinical signs) [21].

After clinical examination, the animals were sedated by intramuscular administration of ketamine hydrochloride (10 mg/kg) plus acepromazine maleate (0.2 mg/kg). Bone marrow was then aspirated from the sternal manubrium. The bone marrow samples were collected into sterile tubes containing EDTA and stored at -20˚C for further analysis by conventional PCR and nested PCR for the detection of *Coxiella burnetii* DNA. Blood (1 to 3 mL) was collected from the cephalic vein into a sterile vacuum tube without anticoagulant. After coagulation, the blood samples were centrifuged at 1125 x *g* and the serum obtained was separated and stored at -20˚C until the time of analysis. The serum was used for the detection of antibodies against *A. platys/A. phagocytophilum*, *B. burgdorferi*, *E. canis/E. ewingii* and *T. gondii* and for the investigation of *D. immitis* antigens. Next, the dogs were culled with an intravenous overdose of sodium thiopental and potassium chloride in accordance with the guidelines of the Federal Council of Veterinary Medicine of Brazil [22], and immediately necropsied.

During necropsy, the organs were examined macroscopically and samples of skin, spleen, liver, lung, heart valves (tricuspid and mitral), uterus, and mammary gland were collected. These tissue samples were fixed in 10% buffered formalin and embedded in paraffin (FFPE) [23] for immunohistochemistry (IHC) (detection of amastigote forms of *Leishmania* spp.) and histopathology. In addition, FFPE samples of the spleen, lungs and mammary glands obtained from *T. gondii*-seropositive dogs in which an inflammatory infiltrate was identified by histopathology were submitted to IHC for the detection of cysts or tachyzoites of this parasite. A second spleen fragment was collected and stored at -20˚C in a sterile RNAse- and DNAse-free tube for subsequent detection of *L. infantum* DNA by singleplex qPCR. For *Leishmania* detection by parasitological culture, samples of skin, spleen, popliteal lymph node and bone marrow were collected aseptically and immersed in sterile saline.

## Histopathology and immunohistochemistry for detection of *Leishmania* and *T. gondii*

Serial sections (5μm) were cut from the paraffin blocks containing the tissues and mounted on non-silanized slides for histopathology and on silanized slides for IHC.

For histopathology, the tissues were stained with hematoxylin-eosin (HE) [23]. The inflammatory infiltrate in the tissues was classified as follows: granulomatous, predominance of cells of the monocyte-macrophage system (activated macrophages, epithelioid macrophages, or multinucleate giant cells); pyogranulomatous, predominance of cells of the monocyte-macrophage system amidst a high number of neutrophils; non-granulomatous, predominance of other types of inflammatory cells (lymphocytes, plasma cells, and neutrophils). In addition, inflammatory cells were quantified using a 1-mm$^2$ optical grid and a manual cell counter under a light microscope. The cells were counted in one field of the HE-stained histological sections at 400× magnification, in the most cellular area of the lesion.

For IHC aimed at detecting amastigote forms of *Leishmania* spp., the tissues were submitted to the steps of deparaffinization, rehydration, blocking of endogenous peroxidase, antigen retrieval, blockade of nonspecific protein binding, and incubation with polyclonal rabbit anti-*Leishmania* serum diluted 1:500, according to the protocol of Boechat et al. [24]. The polymer-based HiDef Detection HRP™ Polymer System (Cell Marque, Rocklin, CA, USA) was used for the detection of amastigote forms of *Leishmania* according to manufacturer recommendations. Histological sections of organs intensely parasitized with amastigote forms of *Leishmania* were incubated with non-immune homologous serum as negative control and with polyclonal rabbit anti-*Leishmania* serum as positive control.

For the evaluation of skin parasite load by IHC, *Leishmania* amastigote forms were quantified as described for the quantification of inflammatory cells. However, the parasites were counted in five fields at 400× magnification in the most parasitized areas. The average number of *Leishmania* amastigote forms was calculated.

For IHC aimed at detecting *T. gondii* cysts and tachyzoites, the tissues (lung, mammary glands, and spleen) were deparaffinized in xylene and rehydrated in decreasing concentrations of ethanol. Endogenous peroxidase was blocked by incubating the histological sections in a solution of 30% hydrogen peroxide and methanol (45 ml of hydrogen peroxide and 55 ml of methanol) for 40 min at room temperature. Antigen retrieval was performed by incubating the histological sections in Declere ™ buffer (Cell Marque, Rocklin, CA, USA) at 110°C in a Decloaking Chamber™ (Biocare Medical, Pacheco, CA, USA). For the blockade of nonspecific protein binding, the histological sections were incubated with Background Block protein blocking solution (Cell Marque, Rocklin, CA, USA) for 10 min at room temperature. The sections were then incubated with polyclonal anti-*T. gondii* antibody (rabbit) (Cell Marque, Rocklin, CA, USA) diluted 1:100 for 1 hour at room temperature. The HiDef Detection HRP™ Polymer System (Cell Marque, Rocklin, CA, USA) was used for the detection of *T. gondii* according to manufacturer recommendations. The enzyme-substrate-chromogen reaction was developed using diaminobenzidine (DAB) plus hydrogen peroxide (Sigma-Aldrich, St. Louis, MO, USA). The sections were counterstained with modified Mayer's hematoxylin (Thermo Scientific, Fremont, CA, USA) for 2 min. Histological sections of tissues intensely parasitized with parasite forms of *T. gondii* were incubated with non-immune homologous serum as negative control and with primary polyclonal anti-*T. gondii* antibody (rabbit) as positive control.

## Parasitological culture and identification of *Leishmania* spp.

The fragments were cultured at 26–28°C in Novy-MacNeal-Nicolle medium plus Schneider's Drosophila medium (Sigma-Aldrich, St. Louis, MO, USA) supplemented with 10% fetal bovine serum and penicillin and streptomycin as antibiotics [25]. The detailed protocol of parasite isolation in culture is registered at https://dx.doi.org/10.17504/protocols.io.22tggen. Parasites isolated in culture were identified as *L. infantum* by multilocus enzyme electrophoresis [26].

## Singleplex qPCR for the diagnosis and quantification of *L. infantum* load

DNA was extracted from the samples using the DNeasy Blood & Tissue kit (Qiagen, Hilden, Germany) according to manufacturer recommendations. Tissue fragments $\leq 10$ mg were used. DNA was quantified in a Qubit 2.0 Fluorometer (Thermo Fisher Scientific, Waltham, MA, USA) using the Qubit dsDNA HS Assay kit (Thermo Fisher Scientific, Waltham, MA, USA) according to manufacturer instructions. Amplification was performed with the StepOne™ System (Applied Biosystems, Foster City, CA, USA) using 0.9 μM of LEISH-1 (5'–AACTTTTCTGGTCCTCCGGGTAG–3') and LEISH-2 (5'-ACCCCCAGTTTCCCGCC3') primers and the TaqMan MGB probe (FAM-5'AAAAATGGGTGCAGAAAT–3'-NFQ), following a previously described protocol [6].

For the quantification of parasite load, a standard curve was constructed with serial dilutions ($10^1$ to $10^5$ parasites) of *L. infantum* DNA (MHOM/BR/1974/PP75). Positive and negative controls were included in each amplification plate and a threshold of 0.1 was established. The DNA of $1 \times 10^5$ promastigote forms of *L. infantum* obtained by parasitological culture was used as positive control and ultrapure water as negative control. Samples in which DNA amplification occurred after the 37th cycle were classified as undetectable. The *L. infantum* load was expressed as the natural logarithm of the number of parasite genome equivalents (gEq)/ng of DNA.

## Conventional PCR for the detection of *C. burnetii* DNA

Forty-five bone marrow samples were submitted to conventional PCR as previously described [27]. The QBT-1 (5'TATGTATCCACCGTAGCCAGT C–3') and QBT-2 (5'–CCCAACAACACCTCCTTATTC–3') primers were used for the amplification of a 687-bp fragment. The reaction mixture contained 0.2 μM of each primer (Invitrogen, Life Technologies, São Paulo, Brazil), 200 μM of dNTP (20 mM of each deoxynucleotide triphosphate), 1.5 mM MgCl$_2$, 0.5 U Platinum® Taq DNA polymerase (Invitrogen, Carlsbad, CA, USA), 4 μL DNA, and nuclease-free water (Promega, Madison, WI, USA) in a final volume of 25 μL. The cycling conditions were described previously [28]. To confirm the amplification, the generated products were separated on an agarose gel stained with 10 μL of Gel RED solution (10,000X) per 100 μL of agarose gel, visualized, and recorded with a photodocumentation system. The target products were then purified using the BigDye Terminator® X-Purification kit (Applied Biosystems, Foster City, CA, USA). The products obtained were sequenced using the BigDye® Terminator V3.1 Cycle Sequencing kit (Applied Biosystems, Foster City, CA, USA) and the ABI PRISM® 3100 software (Applied Biosystems, Foster City, CA, USA). Partial sequences were compared with the htpAB gene from BLAST®.

## Nested PCR for the detection of *C. burnetii* DNA

To increase the sensitivity and specificity of PCR for the detection of *C. burnetii* DNA, all samples were submitted to a second reaction ("nested") using the QBT N3+ (5'-AAG CGT GTG GAG GAG CGA ACC–3') and QBT N4- (5'-CTC GTA ATC ACC AAT CGC TTC GTC–3') primer pair [29]. The positive controls used in PCR were extracted from cultures of cells infected with *C. burnetii*. A volume of the DNA solution of 4 μL was used in the conventional PCR assay and of 2 μL in the nested PCR assay.

For amplification, the reaction contained 1X PCR buffer 10 X, 0.2 μM of each primer (Invitrogen, Life Technologies Brazil), 1.5 mM MgCl$_2$, 200 μM dNTP mixture (20 mM of each deoxynucleotide triphosphate), 0.5 U Platinum Taq DNA polymerase (Invitrogen, Carlsbad, CA, USA), 2 μL of sample DNA, and nuclease-free water (Promega, Madison, WI, USA) in a volume of 25 μL. The amplification was performed in a thermocycler (Applied Biosystem

Veriti 96) using the previously described cycling conditions [29]. To confirm the amplification, the generated products were separated on a 1% agarose gel stained with 10 μL of RED Gel solution (10,000X) per 100 μL of agarose gel. The target products were purified, sequenced, and compared as described in the previous item for conventional PCR.

### Serological detection of antibodies to *Anaplasma* spp., *B. burgdorferi* and *Ehrlichia* spp., and of *D. immitis* antigens

The ELISA 4 Dx® Plus test (IDEXX®, Westbrook, ME, USA) was used for the serological detection of antibodies to *A. platys/A. phagocytophilum*, *B. burgdorferi* and *E. canis/E. ewingii*, and of *D. immitis* antigens following the recommendations of the manufacturer. According to the manufacturer, this test has the following sensitivities and specificities, respectively: 90.3% and 94.3% for *Anaplasma* spp., 94.1% and 96.2% for *B. burgdorferi*, 99.0% and 99.3% for *D. immitis*, and 97.1% and 95.3% for *Ehrlichia* spp.

### Detection of anti-*T. gondii* antibodies in serum

The indirect immunofluorescence antibody test (IFAT) was used as described previously [30]. Anti-dog IgG (whole molecule) FITC antibody produced in rabbits was used as conjugate (Sigma-Aldrich Co., MO, USA). The sera were first diluted 1:16 and then at a ratio of four until 1:256. Samples with a titer of 1:16 or higher were considered positive. Previously known positive and negative controls were included in each analysis. IFAT is considered to have good specificity and sensitivity and is recommended for the immunodiagnosis of *T. gondii* infection in dogs, where it can be used as the only test [31].

### Statistical analysis

Data were analyzed using the free R software, version 3.5.1 [32]. Clinical signs, positivity in the diagnostic tests, and histological changes are described as simple frequencies. For the description of continuous variables (number of inflammatory cells and parasite load), the median (50th percentile) was calculated and the minimum and maximum values are reported. For inflammatory cells, the 75th percentile was also calculated.

The normality of the quantitative variables (parasite load and inflammatory cells) was rejected by the Shapiro-Wilk test at a significance level of 5%, which indicated the use of non-parametric tests for the analysis of these variables. The Mann-Whitney test and Fisher's exact test were used for comparison of the quantitative and qualitative variables, respectively, between the monoinfected and co-seropositive groups. Boxplots were constructed for the graphical presentation of the comparison of parasite load. The median parasite load in the spleen was expressed as the natural logarithm of the parasite genome equivalents/nanogram of DNA (gEq/ng). A p-value $< 0.05$ indicates statistical significance.

### Ethics statement

This study was carried out in strict accordance with the recommendations of the Brazilian Ministry of Health and the Federal Council on Veterinary Medicine, with permission of the owners. The study protocol was approved by the Ethics Committee on Animal Use of the Oswaldo Cruz Foundation (CEUA/Fiocruz; Permit Numbers: LW-54/13 and LW-24/17). The dogs were sedated by intramuscular administration of ketamine hydrochloride and acepromazine maleate and culled with an intravenous overdose of sodium thiopental and potassium chloride. All efforts were made to minimize suffering.

## Results

All 66 *L. infantum*-seropositive dogs investigated in this study had infection with *L. infantum* confirmed by qPCR and/or culture, as described below. Among these 66 dogs, 50 (76%) had antibodies against at least one pathogen other than *L. infantum* (co-seropositive group). The remaining 16 dogs (24%) did not have antibodies against certain pathogens other than *L. infantum* and were thus considered monoinfected with *L. infantum* (monoinfected group). The following frequencies of co-seropositivity for certain pathogens were observed in the 66 dogs investigated: *T. gondii* (59%), *Ehrlichia* spp. (41%), and *Anaplasma* spp. (18%). None of the animals was positive for *B. burgdorferi*, *C. burnetii*, or *D. immitis*. Among the 50 co-sero-positive dogs, 19 (38%) were co-seropositive only for *T. gondii*, 12 (24%) for *T. gondii* and *Ehrlichia* spp., 7 (14%) only for *Ehrlichia* spp., 5 (10%) for *Ehrlichia* spp., *Anaplasma* spp. and *T. gondii*, 3 (6%) for *Ehrlichia* spp. and *Anaplasma* spp., 3 (6%) for *Anaplasma* spp. and *T. gondii*, and 1 (2%) only for *Anaplasma* spp. Statistical analysis of the association of breed with the monoinfected and co-seropositive groups was not possible because of the small number of breed dogs.

Clinical signs compatible with CanL were present in 60 (91%) of the 66 dogs evaluated and absent in six (9%). Clinical signs were observed in 15 (94%) of the 16 dogs monoinfected with *L. infantum* and in 45 (90%) of the 50 co-seropositive dogs. When the frequency of clinical signs of CanL was compared between monoinfected and co-seropositive dogs, no significant difference was observed (p = 0.407). Table 1 shows the frequency of each clinical sign observed in monoinfected and co-seropositive dogs and no significant difference was observed between the two groups.

Evaluation of *L. infantum* load in the spleen revealed positive results in all 66 dogs and a median parasite load of 6.912 gEq/ng. The median *L. infantum* load was 8.589 gEq/ng in the monoinfected group and 6.364 gEq/ng in the co-seropositive group (Fig 1). However, this difference in the median *L. infantum* load in the spleen between the two groups was not significant (p = 0.45). The following median *L. infantum* loads expressed as gEq/ng were found in the spleen of co-seropositive animals: 11.967 when co-seropositive for *Anaplasma* spp. (n = 1), 7.166 (-0.256 to 14.603) when co-seropositive for *T. gondii* (n = 19), 6.340 (-1.808 to 12.054) when co-seropositive for with *Ehrlichia* spp. and *Anaplasma* spp. (n = 3), 5.828 (-1.420 to 16.621) when co-seropositive for *T. gondii* and *Ehrlichia* spp. (n = 12), 5.725 (0.048 to 9.667) when co-seropositive for *Ehrlichia* spp. (n = 7), 5.190 (2.117 to 6.331) when co-seropositive for *Ehrlichia* spp., *Anaplasma* spp. and *T. gondii* (n = 5), and 4.905 (0.986 to 7.705) when co-seropositive for *Anaplasma* spp. and *T. gondii* (n = 3).

Histopathology revealed changes in at least one of the organs studied in 15 (93.8%) of the 16 monoinfected dogs. All 50 (100%) co-seropositive dogs exhibited some type of histological alteration in the examined tissues. Table 2 shows the frequencies of the main histological changes observed in monoinfected and co-seropositive dogs.

The number of inflammatory cells observed in each organ of monoinfected and co-seropositive animals is shown in Table 3. The number of inflammatory cells/mm$^2$ observed in each organ according to the co-seropositivity for certain pathogens is given in the S1 Table.

Immunohistochemistry revealed the presence of amastigote forms of *Leishmania* spp. in all types of organs studied. In addition, amastigote forms were detected in at least one of the organs in 12 (75%) monoinfected dogs and in 39 (78%) co-seropositive dogs. There was no difference in the positivity for amastigote forms of *Leishmania* spp. between monoinfected and co-seropositive dogs in any of the organs studied (Table 4).

The histological changes and amastigote forms detected by IHC in monoinfected and co-seropositive dogs are shown in Fig 2A–2H.

**Table 1. Frequency of clinical signs in dogs infected with *Leishmania infantum*, August 2016 to January 2019 (Barra Mansa, Rio de Janeiro, Brazil).**

| Clinical sign | Monoinfected (n = 16) | Co-seropositive (n = 50) | p-value [a] |
|---|---|---|---|
| Splenomegaly | 10 (62.5%) | 30 (60.0%) | 1.000 |
| Onychogryphosis | 8 (50.0%) | 21 (42.0%) | 0.773 |
| Furfuraceous desquamation of skin | 8 (50.0%) | 21 (42.0%) | 0.238 |
| Dehydration | 7 (43.8%) | 18 (36.0%) | 0.768 |
| Thinness | 6 (37.5%) | 13 (26.0%) | 0.526 |
| Local alopecia | 6 (37.5%) | 12 (24.0%) | 0.340 |
| Regional lymphadenomegaly | 5 (31.3%) | 12 (24.0%) | 0.743 |
| Opaque hair | 6 (37.5%) | 11 (22.0%) | 0.324 |
| Skin ulcers | 3 (18.8%) | 14 (28.0%) | 0.532 |
| Keratoconjunctivitis | 5 (31.3%) | 11 (22.0%) | 0.509 |
| Hepatomegaly | 5 (31.3%) | 10 (20.0%) | 0.493 |
| Apathy | 4 (25.0%) | 9 (18.0%) | 0.719 |
| Pale mucosae | 4 (25.0%) | 9 (18.0%) | 0.719 |
| Generalized alopecia | 3 (18.8%) | 5 (10.0%) | 0.390 |
| Cachexia | 1 (6.3%) | 5 (10.0%) | 1.000 |
| Limb edema | 2 (12.5%) | 2 (4.0%) | 0.245 |
| Generalized lymphadenomegaly | 0 (0.0%) | 4 (8.0%) | -[b] |
| Hyperemic mucosae | 0 (0.0%) | 2 (4.0%) | -[b] |
| Lower limb paresis | 0 (0.0%) | 2 (4.0%) | -[b] |
| Arthralgia | 1 (6.3%) | 0 (0.0%) | -[b] |
| Epistaxis | 0 (0.0%) | 1 (2.0%) | -[b] |
| Jaundice | 0 (0.0%) | 1 (2.0%) | -[b] |
| Pain when palpating the kidneys | 0 (0.0%) | 1 (2.0%) | -[b] |

n: number of dogs. Co-seropositive: *L. infantum*-seropositive dogs with confirmed infection with this parasite by qPCR and/or culture and antibodies against at least one pathogen other than *L. infantum*. Monoinfected: *L. infantum*-seropositive dogs with confirmed infection with this parasite by qPCR and/or culture and without antibodies against certain pathogens other than *L. infantum*.

[a] The p-values were calculated using Fisher's exact test (p ≤ 0.05).

[b] Statistical analysis was not possible because of the absence of the clinical sign in one of the groups.

The median load of amastigote forms of *Leishmania* spp. per $mm^2$ skin quantified by IHC was 2.3 (0–19.6) in monoinfected dogs and 0.9 (0–70.6) in co-seropositive dogs (Fig 3), but the difference was not statistically significant (p = 0.704).

Among the 39 *T. gondii*-seropositive dogs, an inflammatory infiltrate was observed in 33 spleens, 26 lungs, and 14 mammary glands. These samples were processed for IHC to detect *T. gondii* cysts or tachyzoites. One spleen sample and two lung samples were lost during processing. None of the samples examined by IHC were positive for this parasite.

Using parasitological culture for the detection of *Leishmania*, 13 (81.3%) of the monoinfected dogs had positive results in at least one sample studied; 2 dogs (12.5%) did not test positive for *Leishmania* by culture or IHC. Among the co-seropositive dogs, 40 (80%) were positive for *Leishmania* in the parasitological culture, while 5 (10%) did not test positive for *Leishmania* by culture or IHC. In total, 59 dogs (89.4%) tested positive for *Leishmania* by IHC and/or parasitological culture. All parasitological culture isolates from the samples examined were identified as *L. infantum*.

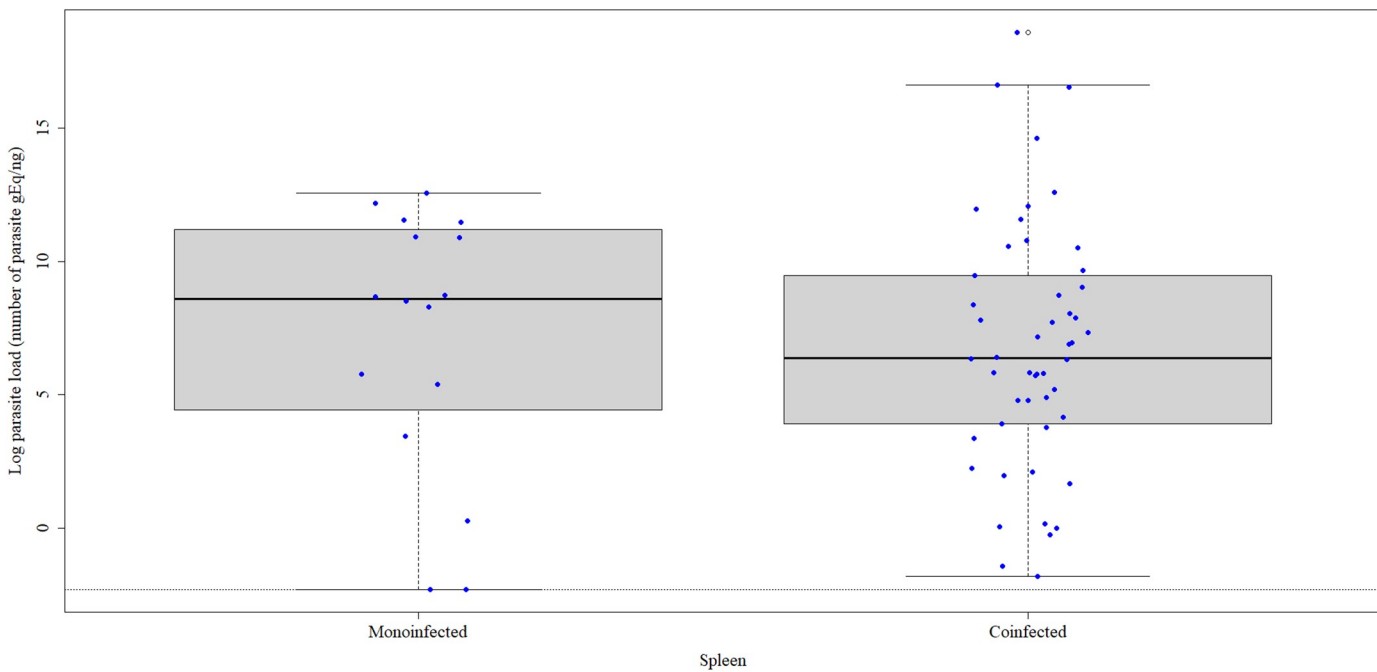

**Fig 1. *Leishmania infantum* load expressed as the median natural logarithm of the number of parasite genomic equivalents (gEq)/ng of DNA in the spleen of monoinfected (n = 16) and co-seropositive (n = 50) dogs.** The horizontal black lines indicate the median parasite load. The vertical dotted lines indicate the interquartile range. The dotted horizontal lines indicate the lower limit of positivity (threshold). The blue dots indicate the parasite load of each dog.

## Discussion

A high frequency of co-seropositivity for certain pathogens in *L. infantum*-seropositive dogs with infection with this parasite confirmed by PCR and/or culture was found, especially co-seropositivity for *T. gondii*. Considering the breed, the results of this study can only be extrapolated to other similar populations composed mainly of mongrel dogs (85%). A high frequency of anti-*T. gondii* (62.9%) antibodies was also detected in dogs from a CanL-endemic area in the state of São Paulo, Brazil [33]. However, in studies conducted in areas not endemic for CanL in Brazil, the authors found frequencies of anti-*Leishmania* and anti-*T. gondii* antibodies of 11.6% in dogs from Londrina, state of Paraná [34], and of 10.2% in dogs from the central region of the state of Rio Grande do Sul [35]. These percentages are lower than those observed in the present study. Paulan et al. [33] found a significant association between reactivity to anti-*L. infantum* and anti-*T. gondii* antibodies in dogs. The authors suggested that the immunosuppression caused by *Leishmania* may increase the susceptibility of these animals with CanL to this coccidium. Additionally, Zulpo et al. [34] observed a higher frequency of anti-*Leishmania* antibodies in dogs seropositive for *T. gondii* and *Neospora*. However, in the present study, it was not possible to assess whether dogs with *L. infantum* are at increased risk of infection with *T. gondii* or vice-versa because of the lack of a control group of dogs not infected with *L. infantum*. The presence of *L. infantum* may have not increased the susceptibility to *T. gondii* infection in this study, as observed in cats [36, 37], since a high frequency of anti-*T. gondii* antibodies has been reported in pigs (65.8%) and chickens (47.8%) in the same area [38]. These results suggest high environmental contamination with *T. gondii* in Barra Mansa, which may explain its high frequency also in dogs.

In the present study, the frequency of *L. infantum* and *Ehrlichia* spp. co-seropositivity was high (41%) and higher than those reported in studies carried out on dogs from Cyprus (36.2%)

**Table 2. Main histological changes in dogs infected with *Leishmania infantum*, August 2016 to January 2019 (Barra Mansa, Rio de Janeiro, Brazil).**

| Sample | Histological changes | Monoinfected (n = 16) | Co-seropositive (n = 50) | p-value[a] |
|---|---|---|---|---|
| Skin (n = 66) | Granulomatous dermatitis | 12 (75.0%) | 34 (68.0%) | 0.758 |
| | Non-granulomatous dermatitis | 0 (0.0%) | 6 (12.0%) | -[d] |
| | Hyperkeratosis | 10 (62.5%) | 28 (56.0%) | 1.000 |
| | Total | 14 (87.5%) | 45 (90.0%) | 0.204 |
| Spleen (n = 66) | Granulomatous splenitis | 7 (43.7%) | 37 (74.0%) | 0.068 |
| | Pyogranulomatous splenitis | 2 (12.5%) | 2 (4.0%) | 0.245 |
| | Non-granulomatous splenitis | 1 (6.2%) | 2 (4.0%) | 1.000 |
| | Total | 10 (62.5%) | 41 (82.0%) | 0.019[c] |
| Liver (n = 66) | Granulomatous hepatitis | 12 (75.0%) | 39 (78.0%) | 1.000 |
| | Pyogranulomatous hepatitis | 2 (12.5%) | 1 (2.0%) | 0.143 |
| | Non-granulomatous hepatitis | 2 (12.5%) | 5 (10.0%) | 1.000 |
| | Vacuolar degeneration of hepatocytes | 9 (56.2%) | 12 (24.0%) | 0.068 |
| | Total | 16 (100.0%) | 46 (92.0%) | 1.000 |
| Lung (n = 66) | Granulomatous pneumonia | 3 (18.7%) | 22 (44.0%) | 0.083 |
| | Pyogranulomatous pneumonia | 3 (18.7%) | 7 (14.0%) | 0.695 |
| | Non-granulomatous pneumonia | 2 (12.5%) | 6 (12.0%) | 1.000 |
| | Total | 8 (50.0%) | 35 (70.0%) | 0.538 |
| Tricuspid (n = 66) | Granulomatous endocarditis | 2 (12.5%) | 7 (14.0%) | 1.000 |
| | Non-granulomatous endocarditis | 1 (6.2%) | 2 (4.0%) | 1.000 |
| | Total | 3 (18.7%) | 9 (18.0%) | 1.000 |
| Mitral (n = 66) | Granulomatous endocarditis | 2 (12.5%) | 3 (6.0%) | 0.588 |
| | Pyogranulomatous endocarditis | 2 (12.5%) | 0 (0.0%) | -[d] |
| | Non-granulomatous endocarditis | 2 (12.5%) | 2 (4.0%) | 0.245 |
| | Total | 6 (37.5%) | 5 (10.0%) | 0.019[c] |
| Mammary gland[b] (n = 28) | Granulomatous mastitis | 4 (44.4%) | 9 (47.4%) | 0.719 |
| | Non-granulomatous endocarditis | 0 (0.0%) | 5 (26.3%) | -[d] |
| | Total | 4 (44.4%) | 14 (73.6%) | 1.000 |
| Uterus[b] (n = 28) | Granulomatous metritis | 0 (0.0%) | 4 (21.0%) | -[d] |
| | Pyogranulomatous metritis | 0 (0.0%) | 1 (5.3%) | -[d] |
| | Non-granulomatous metritis | 0 (0.0%) | 1 (5.3%) | -[d] |
| | Total | 0 (0.0%) | 6 (31.6%) | -[d] |

n: number of dogs. Total: total number of dogs with histological changes in the organ. Co-seropositive: *L. infantum*-seropositive dogs with confirmed infection with this parasite by qPCR and/or culture and with antibodies against at least one pathogen other than *L. infantum*. Monoinfected: *L. infantum*-seropositive dogs with confirmed infection with this parasite by qPCR and/or culture and without antibodies against certain pathogens other than *L. infantum*.

[a] The p-values were calculated using Fisher's exact test (p ≤ 0.05).

[b] The frequencies were calculated and statistical analysis was performed in 9 dogs of the monoinfected group and in 19 dogs of the co-seropositive group.

[c] Statistically significant difference (p ≤ 0.05).

[d] Statistical analysis was not possible because of the absence of certain histological changes in one of the groups.

[39], Côte d'Ivoire (4.9%) [12], and Portugal (2.2%) [40]. This frequency was also higher than those found in dogs from Campo Grande, Mato Grosso do Sul (31.7%) [41], and Cuiabá, Mato Grosso (2.5%) [42], Brazil. On the other hand, higher frequencies of *L. infantum* and *Ehrlichia* spp. co-seropositivity or co-infection than that observed here were reported in Ilha Solteira, São Paulo, Brazil (74.3%) [33], and Catalonia, Spain (55.7%) [7]. Furthermore, the frequency of *L. infantum* and *Anaplasma* spp. co-seropositivity in the present study was higher than that reported in dogs from Cyprus (10.6%) [39] and Portugal (1.1%) [40], but lower than the 52.5%

**Table 3. Number of inflammatory cells/mm² observed in tissues of *Leishmania infantum* monoinfected and co-seropositive dogs, August 2016 to January 2019 (Barra Mansa, state of Rio de Janeiro, Brazil).**

| Samples | Monoinfected (n = 16) | | | | Co-seropositive (n = 50) | | | | p-value[a] |
|---|---|---|---|---|---|---|---|---|---|
| | Min | 50th percentile | 75th percentile | Max | Min | 50th percentile | 75th percentile | Max | |
| Skin (n = 66) | 0 | 195.0 | 243.2 | 481 | 0 | 132.5 | 241.2 | 587 | 0.538 |
| Spleen (n = 66) | 0 | 410.0 | 518.7 | 707 | 0 | 531.5 | 616.7 | 753 | 0.031[c] |
| Liver (n = 66) | 0 | 197.0 | 331.2 | 534 | 0 | 291.5 | 385.0 | 936 | 0.170 |
| Lung (n = 66) | 0 | 103.5 | 326.5 | 434 | 0 | 321.0 | 463.2 | 618 | 0.049[c] |
| Tricuspid (n = 66) | 0 | 0 | 0 | 686 | 0 | 0 | 0 | 300 | 0.888 |
| Mitral (n = 66)[b] | 0 | 0 | 101.5 | 471 | 0 | 0 | 0 | 371 | 0.046[c] |
| Uterus (n = 28)[b] | 0 | 0 | 0 | 0 | 0 | 0 | 127.0 | 544 | -[d] |
| Mammary gland (n = 28)[b] | 67 | 149.0 | 240.0 | 267 | 0 | 323.5 | 427.5 | 581 | 0.041[c] |

n: number of dogs examined; Min: minimum; 50th percentile: median; Max: maximum. Co-seropositive: *L. infantum*-seropositive dogs with confirmed infection with this parasite by qPCR and/or culture and with antibodies against at least one pathogen other than *L. infantum*. Monoinfected: *L. infantum*-seropositive dogs with confirmed infection with this parasite by qPCR and/or culture and without antibodies against certain pathogens other than *L. infantum*.

[a] The p-values were calculated using the Mann-Whitney test (p ≤ 0.05).

[b] The number of inflammatory cells were calculated and statistical analysis was performed in 9 dogs of the monoinfected group and in 19 dogs of the co-seropositive group.

[c] Statistically significant difference (p ≤ 0.05).

[d] Statistical analysis of the organ was not possible because of the absence of inflammatory cells in the monoinfected group.

observed among dogs from Spain [7]. In the United States, frequencies similar to those of the present study were observed in dogs with CanL and co-infected with *Ehrlichia* spp. (41.7%) or *Anaplasma* spp. (41.7%) [10]. Taken together, the results of these surveys and the present findings demonstrate that co-infection with or co-seropositivity for *Ehrlichia* spp. and *Anaplasma* spp. is common among dogs infected with *L. infantum* in different parts of the world.

The frequency of co-infection with or co-seropositivity for *Ehrlichia* spp. and *Anaplasma* spp. can be quite high in regions with favorable environmental conditions for ticks, which are

**Table 4. Frequency of positivity for *Leishmania* amastigotes forms in different organs of dogs infected with *Leishmania infantum* using immunohistochemistry, August 2016 to January 2019 (Barra Mansa, state of Rio de Janeiro, Brazil).**

| Samples | Monoinfected (n = 16) | | Co-seropositive (n = 50) | | p-valor [a] |
|---|---|---|---|---|---|
| | Positive | Negative | Positive | Negative | |
| Skin (n = 66) | 10 (62.5%) | 6 (37.5%) | 29 (58.0%) | 21 (42.0%) | 1.000 |
| Spleen (n = 66) | 9 (56.3%) | 7 (43.7%) | 31 (62.0%) | 19 (38.0%) | 0.772 |
| Liver (n = 66) | 10 (62.5%) | 6 (37.5%) | 30 (60.0%) | 20 (40.0%) | 1.000 |
| Lung (n = 66) | 0 (0.0%) | 16 (100.0%) | 4 (8.0%) | 46 (92.0%) | -[c] |
| Tricuspid (n = 66) | 2 (12.5%) | 14 (87.5%) | 1 (2.0%) | 49 (98.0%) | 0.143 |
| Mitral (n = 66) | 2 (12.5%) | 14 (87.5%) | 0 (0.0%) | 50 (100.0%) | -[c] |
| Uterus (n = 28)[b] | 1 (11.1%) | 8 (88.9%) | 3 (15.8%) | 16 (84.2%) | 1.000 |
| Mammary gland (n = 28)[b] | 5 (55.6%) | 4 (44.4%) | 10 (52.6%) | 9 (47.4%) | 0.173 |

n: number of dogs. Total: total number of dogs with histological changes in the organ. Co-seropositive: *L. infantum*-seropositive dogs with confirmed infection with this parasite by qPCR and/or culture and with antibodies against at least one pathogen other than *L. infantum*. Monoinfected: *L. infantum*-seropositive dogs with confirmed infection with this parasite by qPCR and/or culture and without antibodies against certain pathogens other than *L. infantum*.

[a] The p-values were calculated using Fisher's exact test (p ≤ 0.05).

[b] The frequencies were calculated and statistical analysis was performed in 9 dogs of the monoinfected group and in 19 dogs of the co-seropositive group.

[c] Statistical analysis was not possible because of the absence of positivity in one of the groups.

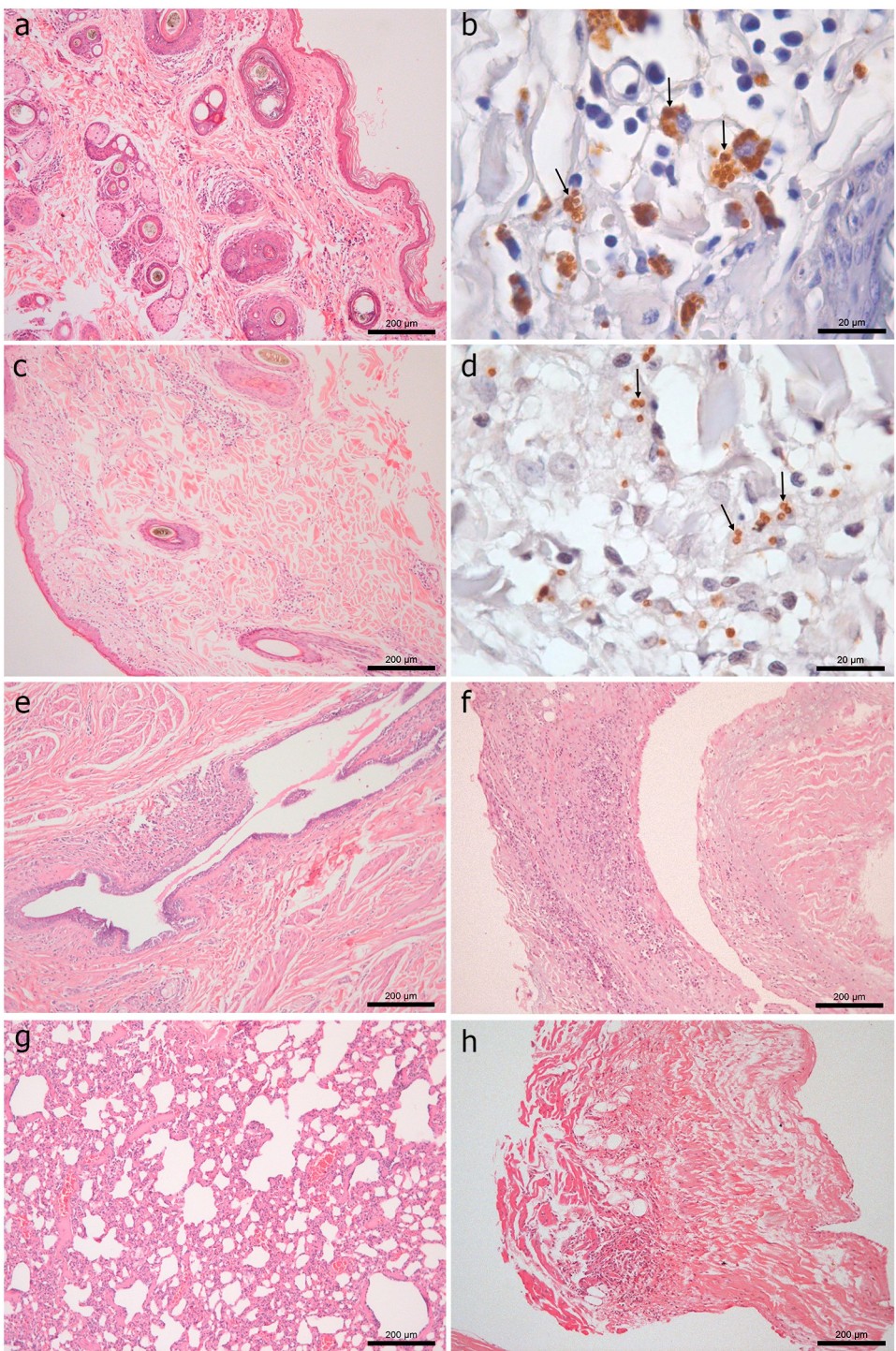

**Fig 2. Histological findings in 66 dogs naturally infected with *Leishmania infantum*.** (A) Hyperkeratosis and granulomatous dermatitis consisting mainly of macrophages, as well as lymphocytes and plasma cells, in a monoinfected dog. HE. (B) Brown-stained amastigote forms of *Leishmania* (arrows) inside macrophages in the skin of the same monoinfected dog. IHC. (C) Diffuse granulomatous dermatitis consisting mainly of macrophages, as well as lymphocytes and plasma cells, in a dog co-seropositive for *Ehrlichia* spp. and *Toxoplasma gondii*. HE. (D) Brown-stained amastigote forms of *Leishmania* (arrows) inside macrophages in the skin of the same dog co-seropositive for *Ehrlichia* spp. and *T. gondii*. IHC. (E) Granulomatous infiltrate around the teat canals and sinuses of the mammary gland in a dog co-seropositive for *T. gondii*. HE. (F) Granulomatous inflammatory infiltrate in the tricuspid valve of a monoinfected dog. HE. (G) Diffuse granulomatous interstitial pneumonia consisting mainly of macrophages in a dog

co-seropositive for *T. gondii*. Note the large numbers of lymphocytes and plasma cells and few neutrophils. HE. (H) Multifocal granulomatous endocarditis and myocarditis consisting mainly of macrophages in a dog co-seropositive for *T. gondii*. Note the large numbers of lymphocytes and few plasma cells. HE.

the vectors of these pathogens [43]. According to Figueredo et al. [43], the presence of ticks and the occurrence of pathogens transmitted by these vectors tend to be more common among dogs in regions where the socioeconomic status of the population is low and dogs are not in close contact with their owners (kept in the backyard or semi-restricted). In the city of Barra Mansa, seropositivity of dogs for the proteobacteria *Rickettsia rickettsii* and/or *R. parkeri*, which are also transmitted by ticks, has been demonstrated, with a frequency of 19.2% [44]. In addition, anti-*R. rickettsii* antibodies and anti-*L. infantum* antibodies were simultaneously detected in 7.7% of the examined dogs [45]. These studies and the present results suggest that dogs from Barra Mansa are frequently exposed to ticks and to the agents transmitted by these arthropods. The investigation of these co-infections or co-seropositivities is important since dogs with CanL can exhibit clinical signs similar to those caused by tick-borne diseases, a fact that makes the diagnosis of *L. infantum* difficult [39, 46]. Consequently, the dog remains in the environment for a longer period and thus serves as a source of *L. infantum* infection for sandflies.

In the present study, the absence of co-seropositivity for *B. burgdorferi*, *C. burnetii* or *D. immitis* in *L. infantum*-seropositive and infected dogs suggests that the circulation of these pathogens is low or does not occur in the area studied. Additional tests such as PCR for the detection of *D. immitis* and *B. burgdorferi* may be useful to identify possible infected and non-seropositive dogs. The absence of *C. burnetii*-positive dogs may be explained by the lack of proximity of the dogs studied with infected sheep and goats, which act as important sources of

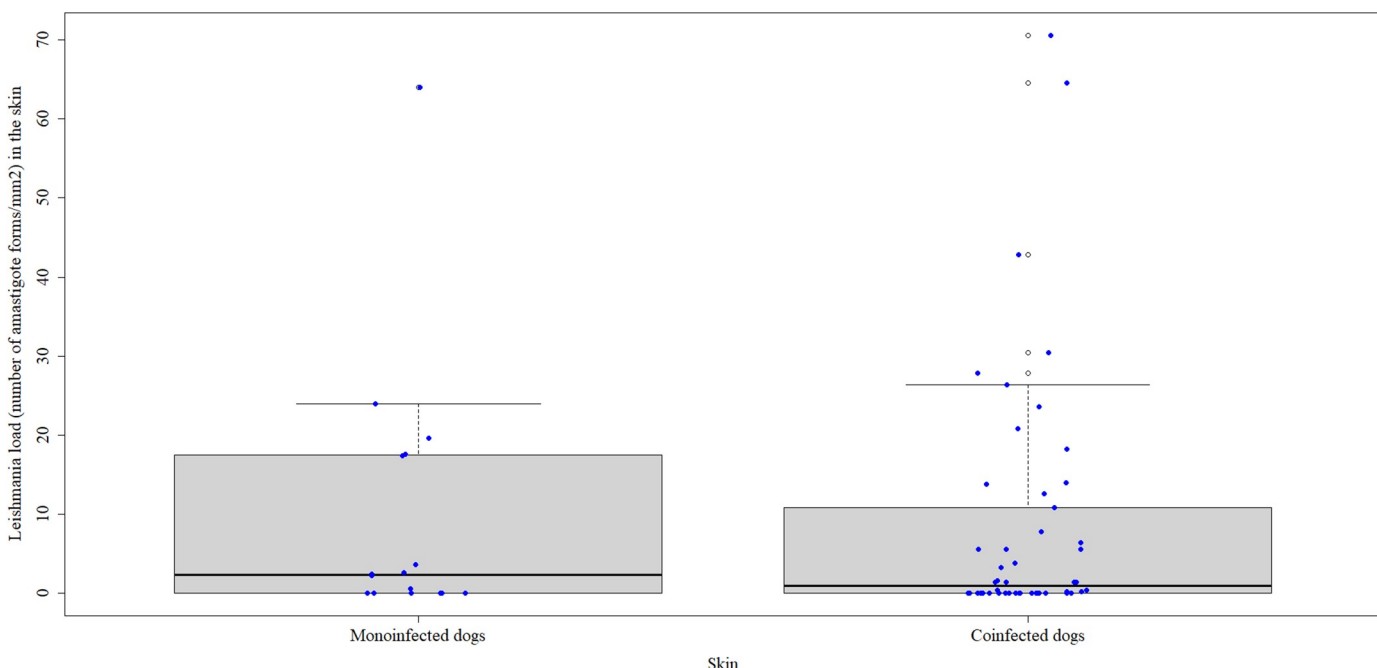

**Fig 3. *Leishmania* load expressed as number of amastigote forms per mm² in the skin of monoinfected and co-seropositive dogs detected by immunohistochemistry.** The horizontal black lines indicate the median parasite load. The vertical dotted lines indicate the interquartile range. The blue dots indicate the parasite load of each dog.

contamination with this bacterium [28]. However, co-infection with *D. immitis* and *L. infan-tum* has been reported in dogs in Brazil [47, 48]. In addition, studies on *L. infantum*-seroposi-tive dogs conducted in Europe demonstrated co-infection with or co-seropositivity for *D. immitis* and *B. burgdorferi*, but the prevalence was low [10, 14, 39, 40, 49–52]. In these surveys, the reported prevalence of *D. immitis* and *L. infantum* co-infection or co-seropositivity in dogs was 1.2% to 8.3% in Portugal [40, 49], 4.3% in Cyprus [39], and 0.16% to 0.6% in Greece [11, 52]. Additionally, clinical cases of this co-infection in three dogs were reported in Italy [50]. Twenty-nine out of 98 (29.6%) dogs were co-infected with *L. infantum* and filariae, including *D. immitis*, in Spain [14]. Co-seropositivity of dogs for *L. infantum* and *B. burgdorferi* was observed in 14 of 2,620 dogs (0.5%) in Greece [11] and in 1 of 1,185 dogs (0.08%) in Portugal [40]. However, this co-seropositivity or co-infection was not found in Cyprus [39] or Côte d'Ivoire [12], in agreement with the present study. Although *C. burnetii*, *D. immitis* and *B. burgdorferi* were not found in the area studied, the detection of *C. burnetii* in two dogs [28] and a prevalence of *D. immitis* of 16.3 to 62.2% [53] and of *B. burgdorferi* of 1.4 to 41.9% [54] have been reported in other cities in the state of Rio de Janeiro. Further studies on the occur-rence of co-infection with these three pathogens in *L. infantum*-seropositive dogs are necessary not only because they are zoonoses, but also because co-infections with filariae have been shown to increase the severity of clinical signs of CanL [14].

In the study of Toepp et al. [10], the risk of *L. infantum* seropositivity was 1.68 times higher among dogs seropositive for agents transmitted by ticks, such as *Ehrlichia* spp. and *Anaplasma* spp., than among dogs that were not co-infected with these agents. In addition, in a study car-ried out in the Campania region in Italy, the presence of antibodies against *Neospora caninum* was the main risk factor for *L. infantum* seropositivity, while the presence of antibodies against *L. infantum* was the main risk factor for *N. caninum* seropositivity [4]. Investigating the occur-rence of co-infections in dogs living in VL-endemic areas is important since *Leishmania* infec-tion can trigger an ineffective immune response that renders dogs more susceptible to other infections, while co-infections can increase the dog's susceptibility to *Leishmania* [4, 7, 10]. These hypotheses are reinforced by the high frequency of co-seropositivities among dogs sero-positive for and/or infected with *L. infantum* found in this study and in other studies [4, 7, 10].

Toepp et al. [10] also observed a significant association between exposure to agents trans-mitted by ticks, such as *Ehrlichia* spp. and *Anaplasma* spp., and the progression of clinical signs of CanL and a higher mortality rate in these dogs. These authors believe that co-infec-tions alter the immunity of dogs, allowing *L. infantum* infection to thrive within the phago-cytes leading to clinical disease. In another study [7], the frequency of anti-*A. phagocytophilum* antibodies was significantly higher in dogs with CanL than in healthy dogs; in addition, these antibodies were more frequent in dogs with more severe clinicopathological changes and with more severe CanL. A study using histopathology found that the percentage of dogs with *L. infantum* amastigote forms on the skin was almost double (36%) in the group of dogs co-infected with *L. infantum* and *E. canis* when compared to the group of *L. infantum*-monoin-fected dogs (19%) [55]. The authors attributed this result to the fact that *E. canis* caused a reduction in the class II histocompatibility complex, leading to depression of the immune sys-tem and favoring the multiplication of *L. infantum*. An experimental study on mice showed that *Ehrlichia* spp. is able to reduce autophagy in infected macrophages, allowing the growth and replication of bacteria and, consequently, of other intracellular agents such as *Leishmania* [56]. However, in the present study, no significant association was found with clinical signs, frequency of positivity for amastigote forms of *Leishmania* spp. in different tissues or *L. infan-tum* load in co-seropositive dogs when compared to monoinfected dogs. This divergence to the literature may be due to the small number of monoinfected animals in the present study, which may have influenced the statistical analysis. Another explanation is the fact that we did

not perform a longitudinal analysis, unlike Toepp et al. [10] who evaluated dogs for the progression of clinical signs and death over a 10-month period. Evaluation of the dogs for the same or a longer period of time than that used by Toepp et al. [10] might have resulted in the observation of worsening of the clinical status and a higher *L. infantum* load in co-seropositive dogs.

Another hypothesis for the lack of aggravation of CanL due to co-seropositivity for certain pathogens in the present study is that the seropositive dogs may not be co-infected or infection with these pathogens is not active, but they were previously exposed to the pathogens and exhibited detectable memory antibodies. Anti-*E. canis* antibodies can persist in the circulation for a long time even after elimination of the agent [57]. In contrast, very recent infections may have also contributed to false-negative results of the serological tests since most of them detect IgG antibodies that are produced later [57]. As a result, dogs classified as monoinfected with *L. infantum* may be infected with the other pathogens investigated but did not have enough time to produce late-phase antibodies against these pathogens.

The possibility of a serological cross-reaction of other pathogens with *L. infantum* must also be taken into account. Other *Leishmania* species [58], *E. canis*, *Babesia canis*, *N. caninum*, *T. gondii* [59], *Leptospira interrogans* [58], *Trypanosoma cruzi* [59], and *T. caninum* [60] can cross-react, thus causing false-positive results in serological *L. infantum* tests. However, previous studies reported that cross-reactivity with *L. infantum* in serological tests for *B. canis* and *E. canis* is rare [61, 62] and the ELISA 4 Dx® Plus test has high sensitivity and specificity. Additionally, in the present study, about 90% of the dogs studied tested positive for *Leishmania* by culture and/or IHC, and all of them tested positive for *L. infantum* by qPCR. Therefore, cross-reactivity with antibodies of other agents is unlikely.

The possibility that the *L. infantum*-monoinfected dogs were actually co-infected with other pathogens not investigated here, such as *Babesia* spp. and helminths, cannot be ruled out. This fact may have contributed to the similarities in clinical signs, *L. infantum* positivity and parasite load between monoinfected and co-seropositive dogs. However, the present results do not rule out the possibility that co-infection with *L. infantum* and other pathogens may exert immunomodulatory activity, preventing a synergistic effect on the aggravation of diseases caused by these agents. Therefore, further studies are needed to better understand the role of co-infections in modulating the immune system of dogs.

Although the number of clinical signs or parasite load was not higher in co-seropositive dogs, a larger number of inflammatory cells was observed in the spleen, lung and mammary gland of these dogs. Studies have associated the inflammatory reaction observed in these organs with *L. infantum* infection in dogs [24, 63]. However, this inflammatory reaction is not a specific histological alteration and it was not possible to correlate its presence with *L. infantum* parasitism in all cases. Nevertheless, the participation of *L. infantum* in these histological changes cannot be ruled based on the lack of detection of amastigote forms of this parasite since these forms are heterogeneously distributed and may not be visualized depending on the histological section examined, even if they are present in the tissue. In addition, even in the absence of amastigote forms, the inflammatory reaction can be triggered by peripheral stimuli such as parasite antigens, DNA or inflammatory mediators, as well as by the deposition of immune complexes in tissues [64, 65]. In a previous study [6], the intensity of the inflammatory infiltrate was higher in the central nervous system of dogs co-infected with *T. gondii* and *E. canis* when compared to dogs infected only with *L. infantum*. Similarly, co-infections may have contributed to aggravate the intensity of the inflammatory infiltrate in the spleen, lung and mammary gland of the co-seropositive dogs studied here. However, the inflammatory infiltrate in these organs was not associated with the detection of *T. gondii* by IHC. This result suggests that *T. gondii* infection was latent in these animals, which is commonly seen in dogs

[66] and was not reactivated by co-infection with *L. infantum*. On the other hand, co-infection with *Ehrlichia* spp. and *Anaplasma* spp. may have contributed to a more intense inflammatory infiltrate in the spleen and lung of the co-seropositive dogs in this study. This can be explained by the fact that a perivascular inflammatory infiltrate consisting of macrophages and lymphocytes is the main histological alteration observed in the lungs and spleen of dogs infected with *E. canis* and *Anaplasma* spp. [67, 68]. These dogs may have been in an early stage of the histological changes associated with *L. infantum* co-infections and the time was not sufficient to develop apparent clinical signs related to the affected organ(s). This hypothesis could explain why co-seropositivity for certain pathogens did not exacerbate the intensity of clinical signs in the dogs studied.

## Conclusions

The present results suggest that co-seropositivities for *Anaplasma* spp., *Ehrlichia* spp. and *T. gondii* are common among dogs infected with *L. infantum* in the region studied. However, these co-seropositivities did not aggravate clinical signs or *L. infantum* load, although they were associated with a more intense inflammatory reaction in some organs.

## Supporting information

**S1 Table. Median number of inflammatory cells observed in each organ of dogs infected with *Leishmania infantum* according to the co-seropositivity for certain pathogens, August 2016 to January 2019 (Barra Mansa, state of Rio de Janeiro, Brazil).**
(DOC)

## Acknowledgments

We thank the Municipal Health Department of Barra Mansa and the Central Laboratory of Public Health (LACEN) for their collaboration; Adilson Benedito de Almeida and Antonio Carlos da Silva from INI, Fiocruz, Igor Falco Arruda from IOC, Fiocruz, and Ricardo Baptista Schmidt from the Oswaldo Cruz Institute (IOC), Fiocruz, for processing the figures.

## Author Contributions

**Conceptualization:** Valéria da Costa Oliveira, Rodrigo Caldas Menezes.

**Data curation:** Valéria da Costa Oliveira, Artur Augusto Velho Mendes Junior, Rodrigo Caldas Menezes.

**Formal analysis:** Valéria da Costa Oliveira, Raquel de Vasconcellos Carvalhaes de Oliveira, Rodrigo Caldas Menezes.

**Funding acquisition:** Fabiano Borges Figueiredo, Rodrigo Caldas Menezes.

**Investigation:** Valéria da Costa Oliveira, Artur Augusto Velho Mendes Junior, Luiz Claudio Ferreira, Tatiana Machado Quinates Calvet, Shanna Araujo dos Santos, Monique Paiva Campos, Francisco das Chagas de Carvalho Rodrigues, Elba Regina Sampaio de Lemos, Tatiana Rozental, Raphael Gomes da Silva, Maria Regina Reis Amendoeira, Rayane Teles-de-Freitas, Rafaela Vieira Bruno, Luciana de Freitas Campos Miranda, Rodrigo Caldas Menezes.

**Methodology:** Valéria da Costa Oliveira, Raquel de Vasconcellos Carvalhaes de Oliveira, Fernanda Nazaré Morgado, Rodrigo Caldas Menezes.

**Project administration:** Valéria da Costa Oliveira, Rodrigo Caldas Menezes.

**Resources:** Fabiano Borges Figueiredo, Rodrigo Caldas Menezes.

**Supervision:** Rodrigo Caldas Menezes.

**Validation:** Valéria da Costa Oliveira, Raquel de Vasconcellos Carvalhaes de Oliveira, Rodrigo Caldas Menezes.

**Visualization:** Valéria da Costa Oliveira, Raquel de Vasconcellos Carvalhaes de Oliveira, Rodrigo Caldas Menezes.

**Writing – original draft:** Valéria da Costa Oliveira, Raquel de Vasconcellos Carvalhaes de Oliveira, Fernanda Nazaré Morgado, Rodrigo Caldas Menezes.

**Writing – review & editing:** Valéria da Costa Oliveira, Artur Augusto Velho Mendes Junior, Luiz Claudio Ferreira, Tatiana Machado Quinates Calvet, Shanna Araujo dos Santos, Fabiano Borges Figueiredo, Monique Paiva Campos, Francisco das Chagas de Carvalho Rodrigues, Raquel de Vasconcellos Carvalhaes de Oliveira, Elba Regina Sampaio de Lemos, Tatiana Rozental, Raphael Gomes da Silva, Maria Regina Reis Amendoeira, Rayane Teles-de-Freitas, Rafaela Vieira Bruno, Fernanda Nazaré Morgado, Luciana de Freitas Campos Miranda, Rodrigo Caldas Menezes.

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
