## [Decision Letter · Decision Letter 0]

30 Dec 2020

PONE-D-20-37262

Frequency of co-infections and their relationship with clinical and histopathological changes and parasite load in dogs infected with Leishmania infantum

PLOS ONE

Dear Dr. Caldas Menezes,

Thank you for submitting your manuscript to PLOS ONE. After careful consideration, we feel that it has merit but does not fully meet PLOS ONE’s publication criteria as it currently stands. Therefore, we invite you to submit a revised version of the manuscript that addresses the points raised during the review process.

A well written manuscript, which deserves consideration. However, the authors should consider to not generalise their results. Also, seropositivity is not equivalent of infection.

We look forward to receiving your revised manuscript.

Kind regards,

Angela Monica Ionica, Ph.D.

Academic Editor

PLOS ONE

Journal Requirements:

Reviewers' comments:

Reviewer's Responses to Questions

**Comments to the Author**

1. Is the manuscript technically sound, and do the data support the conclusions?

Reviewer #1: Yes

Reviewer #2: Yes

2. Has the statistical analysis been performed appropriately and rigorously? 

Reviewer #1: Yes

Reviewer #2: Yes

3. Have the authors made all data underlying the findings in their manuscript fully available?

Reviewer #1: Yes

Reviewer #2: Yes

4. Is the manuscript presented in an intelligible fashion and written in standard English?

Reviewer #1: Yes

Reviewer #2: Yes

5. Review Comments to the Author

Reviewer #1: Under my review criteria, this is a well-written manuscript, which deserves to be considered for publication after the necessary explanations and adaptations as described below. My main concern is whether the number of dogs may be considered as numerically sufficient, together with the hypothesis that these animals might not be representative of dogs from other breeds.

Line 27 – delete “visceral”, as canine leishmaniasis caused by L. infantum is a generalized disease – change accordingly throughout the manuscript

Line 32 – the word “euthanasia” presupposes that the dogs were sick – if this was not the case, use “culling” instead

Line 39 – “co-infected” may not be the right word for all the pathogens; for example, regarding T. gondii, seropositivity may not mean infection – change accordingly (i.e., from co-infected to seropositive) for this and other agents

Line 54 – delete “biological” – including this word suggests that there are also mechanical vectors and this is not the case

Line 59 – whenever possible, display agents alphabetically – change accordingly throughout the manuscript

Line 60 – Isospora is now Cystoisospora; there is more than one species, so please write “Cystoisospora spp.”

Line 64 – replace VL with leishmaniasis

Materials and methods – which criteria have determined a sample size of 66 dogs? Furthermore, this sample composition may not allow extrapolation of results to other breeds.

Line 92 – not clear whether the owners provided signed consent for euthanasia (which is a legal imposition) or to sample collection – please make clearer

Line 112 – abbreviate as T. gondii

Line 174 – Leishmania spp.

Line 181 – L. infantum

Line 234 – Serological detection of antibodies to Anaplasma spp., B. burgdorferi and Ehrlichia spp., and of D. immitis antigens

Line 239 – inform on the per cent sensitivity and specificity for each agent

Line 242 – replace Diagnosis with Detection

Line 247 – which values for this IFAT?

Line 254 – median was calculated, but minimum and maximum were just copied (not calculated)

Line 290 – these dogs may not be infected but only seropositive; please, take this into account in the text

Line 415 – rates are different from percentages; and this is the case of percentages

Discussion – comparison with results from other studies could have included a statistical assessment.

Reviewer #2: It is a very interesting research and very well presented. Please use the term leishmaniosis, instead of leishmaniasis (usually used in human medicine).

My main concern is that you have selected some common and important pathoges to test and if negative you present the dog as a single infection case. However, there are many other important pathogens to be present and contribute. For example sarcoptic mange or worms. I did not see other tests for them. I would suggest to revise the text and title using the phrase certain or common pathogens, which is closer to your work. Also please revise your conclusions accordingly.

6. PLOS authors have the option to publish the peer review history of their article (what does this mean?). If published, this will include your full peer review and any attached files.

Reviewer #1: No

Reviewer #2: No

---

## [Author Response · Author response to Decision Letter 0]

28 Jan 2021

Dear Dr. Caldas Menezes,

Thank you for submitting your manuscript to PLOS ONE. After careful consideration, we feel that it has merit but does not fully meet PLOS ONE’s publication criteria as it currently stands. Therefore, we invite you to submit a revised version of the manuscript that addresses the points raised during the review process.

A well written manuscript, which deserves consideration. However, the authors should consider to not generalise their results. Also, seropositivity is not equivalent of infection.

We look forward to receiving your revised manuscript.

Kind regards,

Angela Monica Ionica, Ph.D.

Academic Editor

PLOS ONE

Journal Requirements:

RESPONSE TO REVIEWER #1

Reviewers' comments:

Reviewer's Responses to Questions

Comments to the Author

Reviewer #1: Under my review criteria, this is a well-written manuscript, which deserves to be considered for publication after the necessary explanations and adaptations as described below. My main concern is whether the number of dogs may be considered as numerically sufficient, together with the hypothesis that these animals might not be representative of dogs from other breeds.

Line 27 – delete “visceral”, as canine leishmaniasis caused by L. infantum is a generalized disease – change accordingly throughout the manuscript

Response:

Line 27. The word “visceral” was deleted and the word “leishmaniasis” was replaced with “leishmaniosis”, as recommended by the reviewer#2. 

In addition, following the recommendations of both reviewers, the word “visceral” was deleted, the word “leishmaniasis” was replaced with “leishmaniosis” and the abbreviation “CVL” was replaced with “CanL” along the text, as indicated below:

Line 101. “canine visceral leishmaniasis (CVL)” was replaced with “ canine leishmaniosis caused by L. infantum (CanL)”

Lines 105, 292, 295, 446, 447, 453, 513, 530, 531 and 549. The abbreviation “CVL” was replaced with the abbreviation “CanL”.

Lines 487 and 527. “VL” was replaced with “CanL”.

Lines 27, 52, 64, 85 and 101. The word “leishmaniasis” was replaced with “leishmaniosis”.

Line 472- “clinical VL caused by caused by L. infantum” was replaced by “CanL”

Line 32 – the word “euthanasia” presupposes that the dogs were sick – if this was not the case, use “culling” instead

Response: 

Lines 33, 92 and 93. The word “euthanasia” was replaced with “culling”.

Lines 91, 117 and 274. The word “euthanized” was replaced with “culled”.

Line 39 – “co-infected” may not be the right word for all the pathogens; for example, regarding T. gondii, seropositivity may not mean infection – change accordingly (i.e., from co-infected to seropositive) for this and other agents

In fact, we can’t prove the infection by the pathogens investigated in this study except for L. infantum. Therefore, the terms “co-infected”, “co-infection” and “co-infections” were replaced with “co-seropositive”, “co-seropositivity” and “co-seropositivities” along all the text, respectively, when refers to the results of this study or to the results of other authors when appropriate. We preferred these terms other than “seropositive”or “seropositivity” for the results of this manuscript, because all the dogs were seropositive for L. infantum. The term “monoinfected” was maintained, because we confirmed the infection with L. infantum by qPCR and/or culture in all the 66 dogs investigated in this study. 

These modifications were done in the following parts of the text:

The title of the manuscript was altered to: “Frequency of co-seropositivities for certain pathogens and their relationship with clinical and histopathological changes and parasite load in dogs infected with Leishmania infantum”

The short title “Co-infections in dogs naturally infected with L. infantum” was replaced with “Co-seropositivities in dogs naturally infected with L. infantum”

The keyword “Co-infection” was replaced with “Co-seropositivity”

Abstract

Lines 28, 29-30. The text “co-infection with other pathogens” was replaced with “co-infections with or co-seropositivities for”.

Line 30. The text “infected with” was replaced with “seropositive for”.

Line 39. The text “co-infected with” was replaced with “co-seropositive for”.

Line 41. The text “co-infected dogs” was replaced with “dogs co-seropositive for certain pathogens”.

Lines 42-43 and 44. The text “co-infected dogs” was replaced with “co-seropositive dogs”.

Line 45. The text “co-infections with L. infantum and other pathogens” was replaced with “infected with L. infantum and co-seropositive for certain pathogens”.

Line 46. The text “ these co-infections” was replaced with” “co-seropositivities for certain pathogens”.

Introduction

Line 58. The text “Co-infections with” was replaced with “Co-infections with or co-seropositivities for”.

Line 67. The text “co-infections with” was replaced with “co-infections with or co-seropositivities for”.

Lines 68-70. The text “co-infected with other pathogens” was replaced with “co-infected with or co-seropositive for certain pathogens”.

Lines 71-72. The text “of co-infections with pathogens” was replaced with” co-infections with or co-seropositivities for certain pathogens”.

Methods

Line 264. The word “co-infected” was replaced with “co-seropositive”.

Results

Lines 283-284. The text “The following pathogens causing co-infections and their frequencies were detected in the 66 dogs investigated:” was replaced with “The following frequencies of co-seropositivity for certain pathogens were observed in the 66 dogs investigated:”

Lines 281, 286, 287, 294, 295, 297, Table 1, 309, 318, 321-326, 330-331, 336, 338, Table 2, 355, 361, 365-366, 370, Table 3, 372, 378, 385-386, Table 4, 392, 398, 401-402, 409, 411, 412, 414-415, 417, 421, 425 and 435. “The word “co-infected” was replaced with “co-seropositive”.

Lines 321-326. The text “ for co-infection with” was replaced with “when co-seropositive for”.

Line 367. The text “coinfecting pathogen” was replaced with “the co-seropositivity for certain pathogens”.

Figures 1 and 3. The word “co-infected” was replaced with “co-seropositive” in the legend. 

Discussion

Lines 442-444. The phrase “A high frequency of seropositivity for pathogens coinfecting dogs naturally infected with L. infantum was found, especially co-infection with T. gondii” was replaced with the “A high frequency of co-seropositivity for certain pathogens in L. infantum seropositive dogs with infection with this parasite confirmed by PCR and/or culture was found, especially co-seropositivity for T. gondii.

Lines 442, 443, 462, 469 and 505. The word “co-infection” was replaced with “co-seropositivity”.

Line 474 and 476. The text “co-infection with” was replaced with “co-infection with or co-seropositivity for”.

Line 482. The text “infection of dogs with” was replaced with “seropositivity of dogs for”.

Lines 483. The text “by serology” was deleted.

Line 487 and 501. The texts “or co-seropositivities” or “or co-seropositivity” were added.

Line 507. The text “this co-infection” was replaced with “this co-seropositivity or co-infection”.

Lines 523-524. The text “frequency of co-infections among dogs with VL found in this and other studies [4, 7, 10].” was replaced with “frequency of co-seropositivities among dogs seropositive for and/or infected with L. infantum found in this study and in other studies [4, 7, 10].

Lines 541, 548, 572, 577, 591 and 596. The word “co-infected” was replaced with “co-seropositive”.

Lines 549-552. The text “Another hypothesis for the lack of aggravation of CVL due to co-infection with other pathogens in the present study was the use of serological techniques for the detection of most of the pathogens investigated. This methodology may have contributed to the classification of exposed animals, but without active infection, as co-infected when, in fact, serological tests may have detected memory antibodies.” was replaced with the text below:

“Another hypothesis for the lack of aggravation of CanL due to co-seropositivity for certain pathogens in the present study is that the seropositive dogs may not be co-infected or infection with these pathogens is not active, but they were previously exposed to the pathogens and exhibited detectable memory antibodies.”

Lines 566 to 568. The phrase “However, 12.5% of monoinfected dogs and 10% of co-infected dogs did not have a positive culture or IHC result, suggesting that infection with L. infantum was not active in these dogs” was deleted.

Line 602. The text “why co-infections” was replaced with “why co-seropositivity for certain pathogens”.

Conclusions

Lines 605 and 607. The word “co-infections” was replaced with “co-seropositivities”.

Other alterations

In addition, due to the alterations in the terms and to make clearer that all the dogs investigated were infected with L. infantum and not only seropositive for this parasite, we reviewed the definition of the groups of dogs of our study, as follows:

Co-seropositive: L. infantum seropositive dogs with confirmed infection with this parasite by qPCR and/or culture and antibodies against at least one pathogen other than L. infantum.

Monoinfected: L. infantum seropositive dogs with confirmed infection with this parasite by qPCR and/or culture and without antibodies against certain pathogens other than L. infantum.

These new definitions of the groups were described in lines 309-312, 355-358, 372-375 and 392-395.

The term “monoinfected” was maintained, because we confirmed the infection with L. infantum by qPCR and/or culture in all the 66 dogs investigated in this study.

The titles of Tables 1, 2, 4 and S1 were also modified as shown below:

Tables 1, 2, 4 and S1. The text “Leishmania infantum-seropositive dogs” was replaced with “dogs infected with Leishmania infantum”.

Table S1. The text “according to the co-infecting pathogen” was replaced with “according to the co-seropositivity for certain pathogens”.

In addition, the following texts were included:

Abstract

Lines 38 to 39. “All 66 dogs tested positive for L. infantum by qPCR and/or culture.”

Results

Lines 279 and 280. “All 66 L. infantum-seropositive dogs investigated in this study had infection with L. infantum confirmed by qPCR and/or culture, as described below.” 

Discussion

Lines 442-443

“…co-seropositivity for certain pathogens in L. infantum-seropositive dogs with infection with this parasite confirmed by PCR and/or culture…”

Line 54 – delete “biological” – including this word suggests that there are also mechanical vectors and this is not the case

Response:

Line 54. The word “biological” was deleted, as recommended.

Line 59 – whenever possible, display agents alphabetically – change accordingly throughout the manuscript

Response:

We desplayed agents alphabetically whenever possible in the following parts of the text:

• Abstract

Lines 33 to 34.

• Introduction

Lines 59 to 64.

• Materials and methods

Lines 115 to 116, 236 to 237 and 239-240.

• Results

Line 286.

• Discussion

Lines 491-492 and 559-560.

• Conclusions

Line 605.

Line 60 – Isospora is now Cystoisospora; there is more than one species, so please write “Cystoisospora spp.”

Line 59. “ Isospora sp.” was replaced with ““Cystoisospora spp.”, as recommended. 

Line 64 – replace VL with leishmaniasis

Response:

VL was replaced with “with leishmaniosis caused by L. infantum”. We replace “leishmaniasis” with “leishmaniosis” along the text, as recommended by the other reviewer (indicated above).

Materials and methods – which criteria have determined a sample size of 66 dogs? 

The sample size included all dogs of the study population, i.e., all dogs that tested seropositive for L. infantum in the town of Barra Mansa and that were sent to INI-Fiocruz during the study period (n=66).

In order to make clearer the criteria for sample size, the following text was included in the materials and methods

Lines 94 to 96. “The sample size included all dogs of the study population, i.e., all dogs that tested seropositive for L. infantum in the town of Barra Mansa and that were sent to INI-Fiocruz during the study period.” 

Furthermore, this sample composition may not allow extrapolation of results to other breeds.

Response:

This is a limitation of the study. Considering the breed, the results of this study can only be extrapolated to other similar populations, composed mainly by mongrel dogs (85%). In addition, statistical analysis of the association of breed with the monoinfected and co-seropositive groups was not possible because of the small number of breed dogs (n=10).

In order to make these limitations clearer in the text, the following texts were included:

Results

Lines 290-291. “Statistical analysis of the association of breed with the monoinfected and co-seropositive groups was not possible because of the small number of breed dogs.”

Discussion

Lines 444-445. Considering the breed, the results of this study can only be extrapolated to other similar populations composed mainly of mongrel dogs (85%).

Line 92 – not clear whether the owners provided signed consent for euthanasia (which is a legal imposition) or to sample collection – please make clearer

Response:

The owners provided signed consent for culling. We clarify this in the line 93.

Line 112 – abbreviate as T. gondii

Response:

Line 116. Toxoplasma gondii was abbreviated as T. gondii.

Line 174 – Leishmania spp.

Response:

Line 177. “Leishmania species” was replaced with “Leishmania spp.”

Line 181 – L. infantum

Response:

Line 184. “Leishmania infantum” was replaced with “L. infantum”.

Line 234 – Serological detection of antibodies to Anaplasma spp., B. burgdorferi and Ehrlichia spp., and of D. immitis antigens

Response:

Lines 236 to 237. The text “Detection of antibodies against Ehrlichia spp., Anaplasma spp. and B. burgdorferi and D. immitis antigens in serum” was replaced with:

 “Serological detection of antibodies to Anaplasma spp., B. burgdorferi and Ehrlichia spp., and of D. immitis antigens”

Lines 238 to 240. The text “...to detect antibodies against E. canis/E. ewingii, A. platys/A. phagocytophilum and Borrelia burgdorferi and antigens of D. immitis…” was replaced with:

“... for the serological detection of antibodies to A. platys/A. phagocytophilum, B. burgdorferi and E. canis/E. ewingii, and of D. immitis antigens...”

Line 239 – inform on the per cent sensitivity and specificity for each agente

Response:

Lines 241-243. The following text was included, as recommended:

“…has the following sensitivities and specificities, respectively: 90.3% and 94.3% for Anaplasma spp., 94.1% and 96.2% for B. burgdorferi, 99.0% and 99.3% for D. immitis, and 97.1% and 95.3% for Ehrlichia spp. 

Line 242 – replace Diagnosis with Detection

Response:

Line 245. “Diagnosis” was replaced with “Detection”.

Line 247 – which values for this IFAT?

Response:

There is no information in the literature about the values of specificity and sensibility of IFAT for the detecion of anti-T. gondii antibodies in the serum of dogs (Dubey et al., 2020; Leal and Coelho, 2014). However, IFAT is considered to have good specificity and sensitivity and is recommended for the immunodiagnosis of T. gondii infection in dogs, where it can be used as the only test (Leal and Coelho, 2014).

References:

Dubey JP, Murata FHA, Cerqueira-Cézar CK, Kwok OCH, Yang Y, SU C. Toxoplasma gondii infections in dogs: 2009-2020. Vet Parasitol. 2020; 287:109223.

Leal PDS, Coelho CD. Toxoplasmosis in dogs: a brief review. Coccidia. 2014; 2: 2-39.

Line 254 – median was calculated, but minimum and maximum were just copied (not calculated)

Response:

Lines 256 to 258. The text “For the description of continuous variables (number of inflammatory cells and parasite load), the median (50th percentile) and the minimum and maximum values were calculated.” was replaced with:”

“For the description of continuous variables (number of inflammatory cells and parasite load), the median (50th percentile) was calculated and the minimum and maximum values are reported.”

Line 290 – these dogs may not be infected but only seropositive; please, take this into account in the text

Response:

The terms “co-infected”, “co-infection” and “coinfections” were replaced with “co-seropositive”, co-seropositivity” and “co-seropositivities”, respectively, along all the text, when refers to the results of this study or to the results of other authors, when appropriate. These alterations are explained in more details above.

Line 415 – rates are different from percentages; and this is the case of percentages

Response:

Line 450. The word “rates” was replaced with “percentages”, as recommended.

Discussion – comparison with results from other studies could have included a statistical assessment.

Response:

This study had a descriptive observational design. Therefore, we chose to compare the findings of our study with the others in the discussion, punctuating whenever possible, the differences in the characteristics of the compared population such as geographical location and study design. To compare specific measurements of our findings with those of other studies using statistical analysis, this study should be designed for systematic review and meta-analysis, with a review carried out with key terms in a systematic way. In addition, detailed results of specific measurements and variability described in the various studies will be necessary, which is outside the scope of this article.

RESPONSE TO REVIEWER #2

Reviewer #2: It is a very interesting research and very well presented. Please use the term leishmaniosis, instead of leishmaniasis (usually used in human medicine).

Response:

 Lines 27, 52, 64, 85, and 101. The term “leishmaniasis” was replaced with”leishmaniosis”.

My main concern is that you have selected some common and important pathoges to test and if negative you present the dog as a single infection case. However, there are many other important pathogens to be present and contribute. For example sarcoptic mange or worms. I did not see other tests for them. I would suggest to revise the text and title using the phrase certain or common pathogens, which is closer to your work. Also please revise your conclusions accordingly.

Response:

The text was revised and the term “pathogens” was replaced with “certain pathogens”, when reffered to the pathogens investigated in this study.

Therefore, the following modifications were done:

The title of the manuscript was altered to: “Frequency of co-seropositivities for certain pathogens and their relationship with clinical and histopathological changes and parasite load in dogs infected with Leishmania infantum”

The word “pathogens” was replaced with “certain pathogens” in the following parts of the text:

Abstract. Lines 30, 41, 45 and 46.

Introduction. Lines 69 and 72.

Results. Lines 282, 284, 312, 358, 363, 367, 375 and 395.

Discussion. Lines 442, 549 and 602.

Legend of S1 Table. Line 816.

---

## [Decision Letter · Decision Letter 1]

10 Feb 2021

Frequency of co-seropositivities for certain pathogens and their relationship with clinical and histopathological changes and parasite load in dogs infected with Leishmania infantum

PONE-D-20-37262R1

Dear Dr. Caldas Menezes,

We’re pleased to inform you that your manuscript has been judged scientifically suitable for publication and will be formally accepted for publication once it meets all outstanding technical requirements.

Kind regards,

Angela Monica Ionica, Ph.D.

Academic Editor

PLOS ONE

Additional Editor Comments (optional):

Reviewers' comments:

Reviewer's Responses to Questions

**Comments to the Author**

1. If the authors have adequately addressed your comments raised in a previous round of review and you feel that this manuscript is now acceptable for publication, you may indicate that here to bypass the “Comments to the Author” section, enter your conflict of interest statement in the “Confidential to Editor” section, and submit your "Accept" recommendation.

Reviewer #1: All comments have been addressed

Reviewer #3: All comments have been addressed

2. Is the manuscript technically sound, and do the data support the conclusions?

Reviewer #1: Yes

Reviewer #3: Yes

3. Has the statistical analysis been performed appropriately and rigorously? 

Reviewer #1: Yes

Reviewer #3: Yes

4. Have the authors made all data underlying the findings in their manuscript fully available?

Reviewer #1: Yes

Reviewer #3: Yes

5. Is the manuscript presented in an intelligible fashion and written in standard English?

Reviewer #1: Yes

Reviewer #3: Yes

6. Review Comments to the Author

Reviewer #1: I am very happy to see that the authors have addressed all my comments and made a substantial effort to accommodate my suggestions.

Reviewer #3: The manuscript PONE-D-20-37262R1 'Frequency of co-seropositivities for certain pathogens and their relationship with clinical and histopathological changes and parasite load in dogs infected with Leishmania infantum' is a well-written and concise manuscript well worth publishing in PLOS ONE. The article presents complex data involving co-infections associated with CanL. The article is very clear and objective and the authors have adequately addressed all comments raised in the previous round of review. This Reviewer has very little else to comment.

7. PLOS authors have the option to publish the peer review history of their article (what does this mean?). If published, this will include your full peer review and any attached files.

Reviewer #1: No

Reviewer #3: No

---

## [Editor Report · Acceptance letter]

22 Feb 2021

PONE-D-20-37262R1 

Frequency of co-seropositivities for certain pathogens and their relationship with clinical and histopathological changes and parasite load in dogs infected with *Leishmania infantum*

Dear Dr. Menezes:

I'm pleased to inform you that your manuscript has been deemed suitable for publication in PLOS ONE. Congratulations! Your manuscript is now with our production department. 

Kind regards, 

on behalf of

Dr. Angela Monica Ionica 

Academic Editor

PLOS ONE